# Accumulated neighbourhood deprivation and coronary heart disease: a nationwide cohort study from Sweden

Sara Larsson Lönn,[1] Olle Melander,[2] Casey Crump,[3] Kristina Sundquist[1]

[1]Center for Primary Health Care Research, Department of Clinical Sciences, Malmö, Lund University, Lund, Sweden
[2]Cardiovascular Research - Hypertension, Department of Clinical Sciences, Malmö, Lund University, Lund, Sweden
[3]Department of Clinical Sciences, Malmö, Icahn School of Medicine at Mount Sinai, New York City, New York, USA

**Correspondence to**
Dr Sara Larsson Lönn;
sara.larsson_lonn@med.lu.se

## ABSTRACT

**Objective** Neighbourhood deprivation is a recognised predictor of coronary heart disease (CHD). The overall aim was to investigate if accumulated exposure to neighbourhood deprivation resulted in higher odds of CHD.
**Design** This is a longitudinal cohort study. Models based on repeated assessments of neighbourhood deprivation as well as single-point-in-time assessments were compared.
**Setting** Sweden.
**Participants** 3 140 657 Swedish men and women without a history of CHD and who had neighbourhood deprivation exposure data over the past 15 years.
**Primary outcome measures** CHD within 5 years' follow-up.
**Results** The results suggested a gradient of stronger association with CHD risk by longer cumulative exposures to neighbourhood deprivation, particularly in the younger age cohorts. Neighbourhood deprivation was also highly correlated over time, especially in older age cohorts.
**Conclusions** The effect of neighbourhood deprivation on CHD might depend on age. Accounting for individuals' baseline age may therefore be important for understanding neighbourhood environmental effects on the development of CHD over time. However, because of high correlation of neighbourhood deprivation over time, single-point-in-time assessments may be adequate for CHD risk prediction especially in older adults.

## INTRODUCTION

Numerous studies have led to the recognition that neighbourhood socioeconomic deprivation is a major determinant of coronary heart disease (CHD).[1–9] However, previous studies of the association between neighbourhood deprivation and CHD have often been cross-sectional or only included a baseline assessment of neighbourhood deprivation, that is, at a single point in time. Conceptual methodological limitations in previous studies include the lack of cumulative measures of neighbourhood exposures; the use of such measures has been suggested as one promising new direction in the research field of neighbourhoods and health.[10] The use of cumulative measures is also in accordance with Hill's criteria,[11] stating that a dose–response association is an important criteria

## Strengths and limitations of this study

► Longitudinal assessments (15 years) of neighbourhood socioeconomic status were conducted, making it possible to assess accumulated exposure to deprived neighbourhoods.
► The study used nationwide register data that are not dependent on self-report.
► There are no lifetime data on neighbourhood exposures.
► As in other studies, this study was unable to identify potentially health-damaging characteristics in the neighbourhood environment that are involved in the development of coronary heart disease.

of a causal relationship. However, even when using a cumulative measure, confounding will most certainly be present in observational studies. Still, the creation of measures of accumulated neighbourhood deprivation based on repeated longitudinal assessments has the potential to take this important research field to the next step. This is in part because CHD develops over a long time period, and longitudinal assessments may therefore represent more accurate measures of the neighbourhood exposure over time in those individuals who develop CHD.

A few previous studies focusing on risk factors for CHD, such as subclinical atherosclerosis and obesity, have been based on repeated, longitudinal assessments of neighbourhood deprivation. Such repeated, longitudinal assessments could be regarded as attempts to construct a dose–response measure of neighbourhood deprivation. For example, trajectory class modelling has been used to identify trajectories of neighbourhood deprivation and their associations with CHD risk factors. One US study used residential history questionnaires to assess trajectory classes of neighbourhood poverty in middle-aged and elderly men and women. Higher cumulative neighbourhood poverty was significantly associated with CHD risk factors

(including subclinical atherosclerosis), particularly in women.[12] Another study, conducted in the UK, found that women who had the longest exposure to neighbourhood deprivation had the greatest weight gain over a period of 10 years.[13] Other studies, focusing on repeated assessments of *individual-level* socioeconomic factors, have shown that repeated exposure to poor individual-level socioeconomic factors increased the risk of subclinical atherosclerosis.[14 15] Neither of these studies, however, assessed the 'hard' outcome CHD, that is, blockage of coronary arteries or myocardial infarction.

When investigating the potential existence of an accumulated 'effect', it is, however, not possible to a priori decide which metric is most suitable for the analysis; instead, it is necessary to analyse various measures and compare how well the models fit the data.[16 17] For example, Mishra *et al*[18] suggest the use of three models to evaluate the accumulation hypothesis. The accumulation hypothesis represents one of several life course approaches in epidemiology that includes the study of long-term effects of different exposures on disease risk later in life.[19]

In this study, the potential effect of accumulated neighbourhood deprivation on CHD was evaluated. We used Swedish nationwide data of men and women aged 45 years and above and who were free from CHD at baseline. The overall aim was to investigate if an accumulated exposure to neighbourhood deprivation resulted in higher risks of CHD. To achieve this aim, we analysed longitudinal assessments of neighbourhood deprivation in addition to a more traditional single-point-in-time assessment. We further investigate whether the results were consistent in different age cohorts and by sex.

## METHODS
### Study sample
We conducted a nationwide cohort study of 3 140 657 Swedish adults (47.5% men) with information on neighbourhood deprivation each year during 15 years of potential exposure (see the Measures section) and no registered CHD prior to baseline. Baseline was the year the individual turned 45, 50, 55, 60, 65, 70, 75 or 80. To attain coverage in the medical registers that was comparable between study subjects and avoid inclusion of individuals in more than one cohort, we only included those who attained their 'baseline age' (ie, 45, 55, 65, 70 and so on) between 2003 and 2007. We linked several nationwide Swedish registers (see below) using the unique 10-digit personal identification number, which is assigned at birth or immigration to all permanent residents in Sweden. Each personal identification number was replaced with a serial number to ensure confidentiality of all individuals. Together with the geographical data, the following data sources were used to create our data set: the Total Population Register, containing information about year of birth, sex and marital status; the Longitudinal Integration Database for Health Insurance and Labor Market Studies, including annual information on income, employment,

social welfare and education; the Hospital Discharge Register, containing hospitalisations; the Out-patient Care Register, containing information from all outpatient clinics; and the Mortality Register with dates and causes of death. We stratified the analysis by age cohort and sex.

### Patient and public involvement
The study was based on secondary data. No patients were involved in setting the research question or the outcome measures, nor were they involved in developing plans for design or implementation of the study. No patients were asked to advise on interpretation or writing up of results. The results will be disseminated to patients and the public through a website and press releases suitable for a non-specialised audience.

### Measures
The outcome variable was CHD within 5 years after baseline. We identified the first CHD event in each individual from Swedish Medical Registers based on the codes from WHO's International Classification of Diseases (ICD), that is, ICD-7 code 420, ICD-8 and ICD-9 codes 410, 411, 412, 413 and 414, and ICD-10 codes I20, I21, I22, I23, I24 and I25. Those who died during the 5-year follow-up were censored at the time of death.

The exposure variable, neighbourhood deprivation, was based on Small Areas for Market Statistics (SAMS) obtained from Statistics Sweden, the Swedish government-owned statistics bureau. There are approximately 9200 SAMS throughout Sweden, with an average population of around 1000 inhabitants. The SAMS units are relatively small and, in qualitative studies, small neighbourhoods have been shown to be consistent with how residents themselves define their neighbourhoods.[20] We assessed the socioeconomic characteristics of each neighbourhood using an aggregated measure based on four dimensions of deprivation in the working population aged 25–64 (as these individuals are more socioeconomically active than young adults and retirees), namely the proportion of people residing in the neighbourhood with low income, low education, unemployment and receipt of social welfare. The neighbourhood deprivation measure, which has been described elsewhere, is a weighted score of the four dimensions described above.[21] The aggregated measure was standardised to have a mean of 0 and SD of 1 each year, making it a relative measure comparable between years. A highly deprived neighbourhood was defined as a neighbourhood with a deprivation score over 1, and an affluent neighbourhood (ie, low neighbourhood deprivation) was defined as a neighbourhood with a deprivation score under −1.

The exposure neighbourhood variables used in the analyses were based either on a single-point-in-time assessment, assessed the year before baseline, or repeated assessments from the 15 years prior to baseline, divided into three 5-year periods. For the single-point-in-time measure, we used three exposure categories, that is, high, middle and low neighbourhood deprivation, while for

the accumulated exposure we constructed a composite measure based on the 15 years prior to baseline. This means that the 15 years of exposure depends on the year each individual reaches their 'baseline age'. When creating our accumulated exposure variable, we first assessed whether the individuals had lived in a deprived neighbourhood at any time in each of the three 5-year periods and constructed a more informative variable defined by eight patterns of longitudinal exposure, including (0,0,0), representing never exposed; (1,0,0), (0,1,0) and (0,0,1), representing exposure in one of the three 5-year periods with the number 1 indicating in which of the three periods prior to baseline the exposure occurred, that is, 11–15, 6–10 or 1–5 years before baseline; (1,1,0), (1,0,1) and (0,1,1), representing exposure during two of the three 5-year periods; and (1,1,1), representing exposure during all three 5-year periods. Our accumulated exposure variable is a composite measure of these eight categories where the exposure is independent of time, that is, one 5-year period of exposure, two 5-year periods of exposure or exposure in all three 5-year periods. Other individual-level variables were assessed at baseline and included to adjust for confounding. As measures of individual socioeconomic status, we used education and income. Education was categorised into low (elementary school only), middle (more than elementary school but no university studies) and high (university studies). Missing information was treated as having low education. This was the case for 0.1% of the Swedish-born study population and for 0.5% of the foreign-born study population. Income was defined in each age cohort by the family-adjusted income and categorised into quartiles. For marital status, we used four categories: unmarried, married, divorced and widowed. Psychiatric disorder was defined as having a pre-existing main diagnosis in the Hospital Discharge Register based on the following codes: ICD-8: 29 and 30; ICD-9: 311–314 and 316; and ICD-10: F0-F6 and F9. This variable was included as it is a known confounder of CHD and neighbourhood deprivation.[22 23]

## Statistical analyses

To increase the understanding of our neighbourhood deprivation measure, we estimated pairwise tetrachoric correlations between the 5-year periods (period 2 vs 1, period 3 vs 2, and period 3 vs 1) in each age cohort.

We analysed the association between neighbourhood deprivation and CHD within 5 years after baseline using logistic regression with different measures of exposure to neighbourhood deprivation, either as a single-point-in-time measure at baseline or as an aggregated measure of the 15 years prior to baseline. To account for potential confounding, we adjusted for education, marital status, income and psychiatric hospitalisation. Results are presented as ORs with 95% CIs. First, we fitted the model based on a single-point-in-time measure including three exposure categories: low, middle or high neighbourhood deprivation, treated as a categorical variable (model 1, *single-point-in-time model*). Second, we analysed

an accumulated model, based on the three composite exposure periods, representing one ((1,0,0), (0,1,0) or (0,0,1)), two ((1,1,0), (1,0,1) or (0,1,1)), or three (1,1,1) periods of exposure and compared with the category never exposed ((0,0,0)) (model 2, *categorical accumulated model*). This model predicts CHD as a function of the number of exposed periods without considering the timing of the exposure. In a first sensitivity analysis, we used a continuous accumulated model, where the number of exposed 5-year periods was included as a continuous variable (model S2a, *continuous accumulated model, 5-year*). This model represents a scenario where we assume that each exposed period has the same impact on the increase in odds. In a second sensitivity analysis, we constructed a model using all eight categories of longitudinal assessments as exposure variable to explore on the possible effect of timing (model S2b, *timing/period model*). This model predicts CHD as a function of timing and number of exposed periods. Comparing model S2b with model 2 evaluates if it is reasonable to summarise the number of exposed periods without considering the timing of exposure. Finally, we conducted a third sensitivity analysis where we constructed an additional continuous accumulation model, where the number of exposed *1-year* periods was included as a continuous variable (model S2c, *continuous accumulated model, 1-year*). The equations and description of all these models can be found in online supplementary table 1. We compared the models using the Akaike information criterion (AIC) as a measure of model fit, where a lower value indicates a better fit after taking the number of included variables into account. In addition, we also used a fixed deprivation measure, from the year 2000, so that neighbourhoods could not change ranking over time, to investigate how this would affect the estimated ORs. All statistical analyses were performed in SAS V.9.3 in the SAS system for Windows.

## Summary of statistical models

- ► Model 1: single-point-in-time model.
- ► Model 2: categorical accumulated model.
- ► Model S2a: sensitivity analysis A, continuous accumulated model (5-year).
- ► Model S2b: sensitivity analysis B, timing/period model.
- ► Model S2c: sensitivity analysis C, continuous accumulated model (1-year).

## RESULTS

Table 1 a,b shows the sample size and cumulative 5-year incidence of CHD for men and women by neighbourhood exposure category and age cohort. Higher cumulative 5-year incidence was found in the older age cohorts (compared with the younger) and in men (compared with women). Depending on neighbourhood exposure category, the cumulative incidence of CHD in men ranged from 1%–2% in the age cohort 45–49 years at baseline to 15%–16% in the age cohort 80–84 years at baseline. The corresponding cumulative incidence for women was

**Table 1a** Total number and cumulative 5-year incidence of CHD events in men

| Category | Never exposed | One period of exposure | Two periods of exposure | Three periods of exposure |
|---|---|---|---|---|
| No CHD at 45 | 114 844 | 28 865 | 23 316 | 28 284 |
| Deaths 45–49 | 730 (0.64%) | 231 (0.8%) | 221 (0.95%) | 337 (1.19%) |
| CHD 45–49 | 1211 (1.05%) | 365 (1.26%) | 321 (1.38%) | **468 (1.65%)** |
| No CHD at 50 | 154 223 | 30 647 | 25 144 | 34 452 |
| Deaths 50–54 | 1525 (0.99%) | 444 (1.45%) | 423 (1.68%) | 686 (1.99%) |
| CHD 50–54 | 2989 (1.94%) | 797 (2.60%) | 699 (2.78%) | **1004 (2.91%)** |
| No CHD at 55 | 167 584 | 29 780 | 24 712 | 34 132 |
| Deaths 55–59 | 2801 (1.67%) | 699 (2.35%) | 665 (2.69%) | 1001 (2.93%) |
| CHD 55–59 | 4292 (2.56%) | 936 (3.14%) | 814 (3.29%) | **1210 (3.55%)** |
| No CHD at 60 | 179 878 | 28 188 | 23 173 | 33 535 |
| Deaths 60–64 | 5027 (2.79%) | 961 (3.41%) | 936 (4.04%) | 1546 (4.61%) |
| CHD 60–64 | 8874 (4.93%) | 1598 (5.67%) | 1454 (6.27%) | **2173 (6.48%)** |
| No CHD at 65 | 128 389 | 19 462 | 16 058 | 25 585 |
| Deaths 65–69 | 5959 (4.64%) | 1152 (5.92%) | 1010 (6.29%) | 1838 (7.18%) |
| CHD 65–69 | 7032 (5.48%) | 1190 (6.11%) | 1002 (6.24%) | **1708 (6.68%)** |
| No CHD at 70 | 93 675 | 14 764 | 12 505 | 20 259 |
| Deaths 70–74 | 7519 (8.03%) | 1449 (9.81%) | 1222 (9.77%) | 2392 (11.81%) |
| CHD 70–74 | 8710 (9.30%) | 1490 (10.09%) | 1224 (9.79%) | **2313 (11.42%)** |
| No CHD at 75 | 72 900 | 12 061 | 10 562 | 17 393 |
| Deaths 75–79 | 10 171 (13.95%) | 1981 (16.42%) | 1823 (17.26%) | 3038 (17.47%) |
| CHD 75–79 | 7076 (9.71%) | 1287 (10.67%) | 1070 (10.13%) | **1943 (11.17%)** |
| No CHD at 80 | 55 884 | 9478 | 7908 | 14 272 |
| Deaths 80–84 | 13 843 (24.77%) | 2667 (28.14%) | 2193 (27.73%) | 4024 (28.2%) |
| CHD 80–84 | 8436 (15.10%) | **1552 (16.37%)** | 1248 (15.78%) | 2321 (16.26%) |

**Table 1b** Total number and cumulative 5-year incidence of CHD events in women

| Category | Never exposed | One period of exposure | Two periods of exposure | Three periods of exposure |
|---|---|---|---|---|
| No CHD at 45 | 118 354 | 27 389 | 21 137 | 25 903 |
| Deaths 45–49 | 521 (0.44%) | 144 (0.53%) | 146 (0.69%) | 185 (0.71%) |
| CHD 45–49 | 602 (0.51%) | 173 (0.63%) | 122 (0.58%) | **226 (0.87%)** |
| No CHD at 50 | 159 942 | 30 356 | 24 972 | 32 526 |
| Deaths 50–54 | 1262 (0.79%) | 296 (0.98%) | 287 (1.15%) | 405 (1.25%) |
| CHD 50–54 | 1379 (0.86%) | 337 (1.11%) | 332 (1.33%) | **490 (1.51%)** |
| No CHD at 55 | 173 835 | 29 434 | 24 873 | 33 326 |
| Deaths 55–59 | 2050 (1.18%) | 453 (1.54%) | 437 (1.76%) | 656 (1.97%) |
| CHD 55–59 | 1829 (1.05%) | 437 (1.48%) | 376 (1.51%) | **582 (1.75%)** |
| No CHD at 60 | 186 457 | 28 658 | 24 223 | 34 919 |
| Deaths 60–64 | 3667 (1.97%) | 717 (2.5%) | 691 (2.85%) | 1113 (3.19%) |
| CHD 60–64 | 3999 (2.14%) | 808 (2.82%) | 741 (3.06%) | **1140 (3.26%)** |
| No CHD at 65 | 138 979 | 21 478 | 17 852 | 28 714 |
| Deaths 65–69 | 4306 (3.1%) | 809 (3.77%) | 707 (3.96%) | 1341 (4.67%) |
| CHD 65–69 | 3774 (2.72%) | 705 (3.28%) | 601 (3.37%) | **1116 (3.89%)** |
| No CHD at 70 | 110 552 | 18 147 | 15 300 | 25 782 |
| Deaths 70–74 | 5885 (5.32%) | 1146 (6.32%) | 950 (6.21%) | 1962 (7.61%) |
| CHD 70–74 | 5694 (5.15%) | 1172 (6.46%) | **1003 (6.56%)** | 1637 (6.35%) |
| No CHD at 75 | 99 419 | 17 453 | 14 454 | 25 731 |

**Table 1b** Continued

| Category | Never exposed | One period of exposure | Two periods of exposure | Three periods of exposure |
|---|---|---|---|---|
| Deaths 75–79 | 9225 (9.28%) | 1838 (10.53%) | 1710 (11.83%) | 3055 (11.87%) |
| CHD 75–79 | 5964 (6.00%) | 1217 (6.97%) | **1043 (7.22%)** | 1820 (7.07%) |
| No CHD at 80 | 86 498 | 15 113 | 13 217 | 23 752 |
| Deaths 80–84 | 15 114 (17.47%) | 2921 (19.33%) | 2549 (19.29%) | 4604 (19.38%) |
| CHD 80–84 | 9212 (10.65%) | 1731 (11.45%) | 1560 (11.80%) | **2848 (11.99%)** |

The highest cumulative incidence for each age cohort is in bold.
CHD, coronary heart disease.

0.5%–0.9% in the age cohort 45–49 years at baseline and 11%–12% in the age cohort 80–84 years at baseline. For men, the neighbourhood exposure categories with the highest cumulative incidence in each age stratum were, with one exception, in the three-period category (marked in bold). For women, the pattern was similar to the one in men; the cumulative incidence was, in six out of eight cohorts, highest in the three-period category. For both men and women, these deviations were found in the elderly, where the relative risk increase due to accumulated exposure was less pronounced.

In all age groups and in both sexes, the lowest cumulative incidence of CHD was, with only a few exceptions, found among those men and women who had not lived in a deprived neighbourhood at any time during the 15-year assessment period.

### Correlations between time periods

The tetrachoric correlations for the neighbourhood deprivation measure between the different time periods for each age cohort are shown in table 2. For both men and women in all age cohorts, the correlations between different time periods were higher for periods closer in time. For both men and women, the lowest correlations were found between the two 5-year periods that were most separated in time, that is, 11–15 years vs 1–5 years before baseline, and in the youngest age cohort (0.68). The correlations between time periods increased with age, and the highest correlations were found when comparing the period 6–10 years with the period 1–5 years before baseline in the oldest age cohort for both men and women (0.92).

### Single-point-in-time assessment (model 1)

The adjusted ORs with 95% CIs, obtained from model 1, are presented, by sex and by age cohort in figure 1. The corresponding estimates for all models can be found in online supplementary table 2a,b.

The reference category represents individuals living in the least deprived (ie, most affluent) neighbourhoods. For men, all age cohorts living in the most deprived neighbourhoods had higher odds for CHD than those living in the least deprived neighbourhoods, with ORs ranging from 1.12 (95% CI 1.05 to 1.20) to 1.42 (95% CI 1.29 to 1.57) (figure 1). In most age cohorts among

men, the odds for CHD among those living in neighbourhoods with a middle level of neighbourhood deprivation were also higher than for those living in the least deprived neighbourhoods, with ORs ranging from 1.03 (95% CI 0.97 to 1.10) to 1.18 (95% CI 1.09 to 1.28). A similar pattern was found in women, although the ORs were slightly higher than in men, ranging from 1.20 (95% CI 1.12 to 1.28) to 1.56 (95% CI 1.26 to 1.92) for women in the most deprived neighbourhoods and from 1.10 (95% CI 1.04 to 1.17) to 1.28 (95% CI 1.15 to 1.42) (figure 1) for women living in neighbourhoods with a middle level of neighbourhood deprivation. In general, the magnitude of the ORs was lower in the older cohorts,

**Table 2** Tetrachoric correlations (SE) of exposure to neighbourhood deprivation between 5-year periods

| | Period 2 vs 1 | Period 3 vs 2 | Period 3 vs 1 |
|---|---|---|---|
| **Men** | | | |
| No CHD at 45 | 0.833 (0.002) | 0.856 (0.002) | 0.677 (0.003) |
| No CHD at 50 | 0.861 (0.001) | 0.885 (0.001) | 0.729 (0.002) |
| No CHD at 55 | 0.871 (0.001) | 0.892 (0.001) | 0.742 (0.002) |
| No CHD at 60 | 0.882 (0.001) | 0.903 (0.001) | 0.767 (0.002) |
| No CHD at 65 | 0.892 (0.001) | 0.912 (0.001) | 0.785 (0.002) |
| No CHD at 70 | 0.891 (0.001) | 0.912 (0.001) | 0.782 (0.002) |
| No CHD at 75 | 0.896 (0.001) | 0.911 (0.001) | 0.782 (0.002) |
| No CHD at 80 | 0.899 (0.001) | 0.915 (0.001) | 0.788 (0.002) |
| **Women** | | | |
| No CHD at 45 | 0.833 (0.002) | 0.865 (0.001) | 0.682 (0.003) |
| No CHD at 50 | 0.854 (0.001) | 0.884 (0.001) | 0.721 (0.002) |
| No CHD at 55 | 0.869 (0.001) | 0.894 (0.001) | 0.738 (0.002) |
| No CHD at 60 | 0.883 (0.001) | 0.904 (0.001) | 0.765 (0.002) |
| No CHD at 65 | 0.891 (0.001) | 0.914 (0.001) | 0.782 (0.002) |
| No CHD at 70 | 0.889 (0.001) | 0.914 (0.001) | 0.780 (0.002) |
| No CHD at 75 | 0.892 (0.001) | 0.912 (0.001) | 0.781 (0.002) |
| No CHD at 80 | 0.895 (0.001) | 0.915 (0.001) | 0.784 (0.002) |

Period 1 refers to 11–15 years prior to baseline, period 2 to 5–10 years prior, and period 3 to 1–5 years prior.
CHD, coronary heart disease.

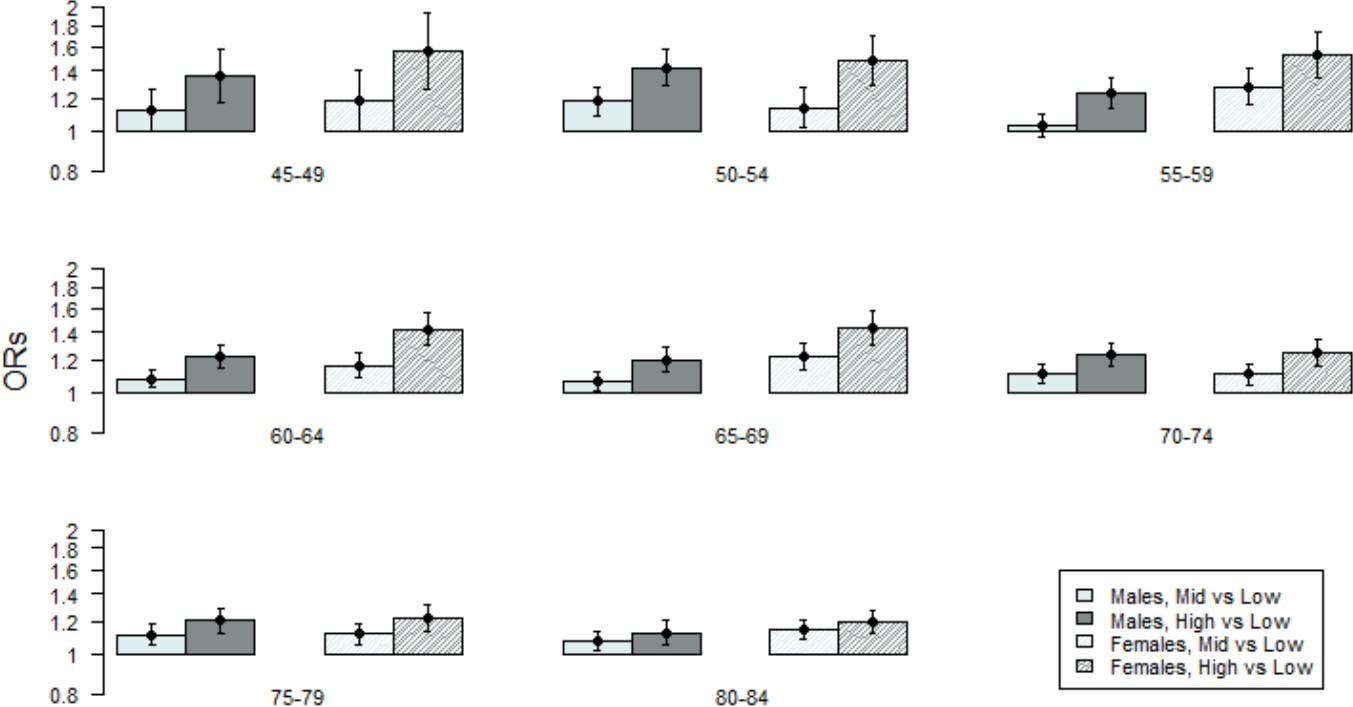

**Figure 1** Adjusted ORs and 95% CIs (on a logarithmic scale) representing the association between neighbourhood deprivation category and coronary heart disease using the single-point-in time model (model 1) in different age cohorts.

probably driven by the higher overall cumulative incidences resulting in lower relative odds.

### Accumulated assessments (models 2, S2a, S2b and S2c)

The adjusted ORs and 95% CIs, obtained from model 2, are presented, by sex and age cohort, in figure 2. The corresponding estimates for all models can be found in online supplementary table 3 a,b, together with the estimates from our sensitivity analyses (models S2a, S2b and S2c, which are found in online supplementary tables 3 a,b and 4 a,b). Exposure to three, two or one time period in a deprived neighbourhood was compared with no exposure. Between ages 45 years and 79 years in men and between ages 45 years and 69 years in women, those in the three time periods' exposure category had the strongest associations with CHD, ranging from 1.11 (95% CI 1.06 to 1.18) to 1.30 (95% CI 1.16 to 1.45) (figure 2). In addition, for men up to age 69 years, there was a trend where two time periods of exposure were associated with a higher odds of CHD, ranging from 1.07 (95% CI 1.00 to 1.14) to 1.28 (95% CI 1.17 to 1.39), then one period, which was associated with increased ORs ranging from 1.06 (95% CI 0.99 to 1.13) to 1.23 (95% CI 1.14 to 1.33). This trend was also observed in women, although less pronounced than in men. Three periods of exposure showed the strongest association up to age 69, ranging from 1.29 (95% CI 1.20 to 1.38) to 1.48 (95% CI 1.33 to 1.65), and two periods showed a stronger association than one period in three out of these four cohorts, ranging from 1.14 (95% CI 1.04

to 1.25) to 1.34 (95% CI 1.19 to 1.52) (figure 2). One period of exposure resulted in increased ORs ranging from 1.06 (95% CI 1.01 to 1.12) to 1.25 (95% CI 1.12 to 1.39). At older ages, there were only minor differences between the exposure categories. The sensitivity analysis, based on all eight exposure categories (model S2b, timing/period model; online supplementary table 3 a,b), suggests that the categorical accumulative model is useful for the younger cohorts of men and women and that adding information of the timing of exposure is not necessary, based on the AIC values (ie, lower for model 2 compared with model S2b). The two continuous accumulated models (used for the two other sensitivity analyses) using the number of 5-year periods (model S2a) or 1-year periods (model S2c) also suggest that the associations were stronger for younger cohorts of men and women.

As suggested above, the weaker associations observed in the older age cohorts may partly be a result of the relatively higher overall incidence rates in the older age cohorts.

Up to age 64 years, the categorical accumulated model (model 2) provided a better fit to the data (lower AIC values) in all four of the male cohorts and in three out of the four female cohorts compared with the single-point-in-time model (model 1). After the age of 65 there was no clear pattern, although the difference between the two models was minor, suggesting that the single-point-in-time measure is a valid approximation of the neighbourhood

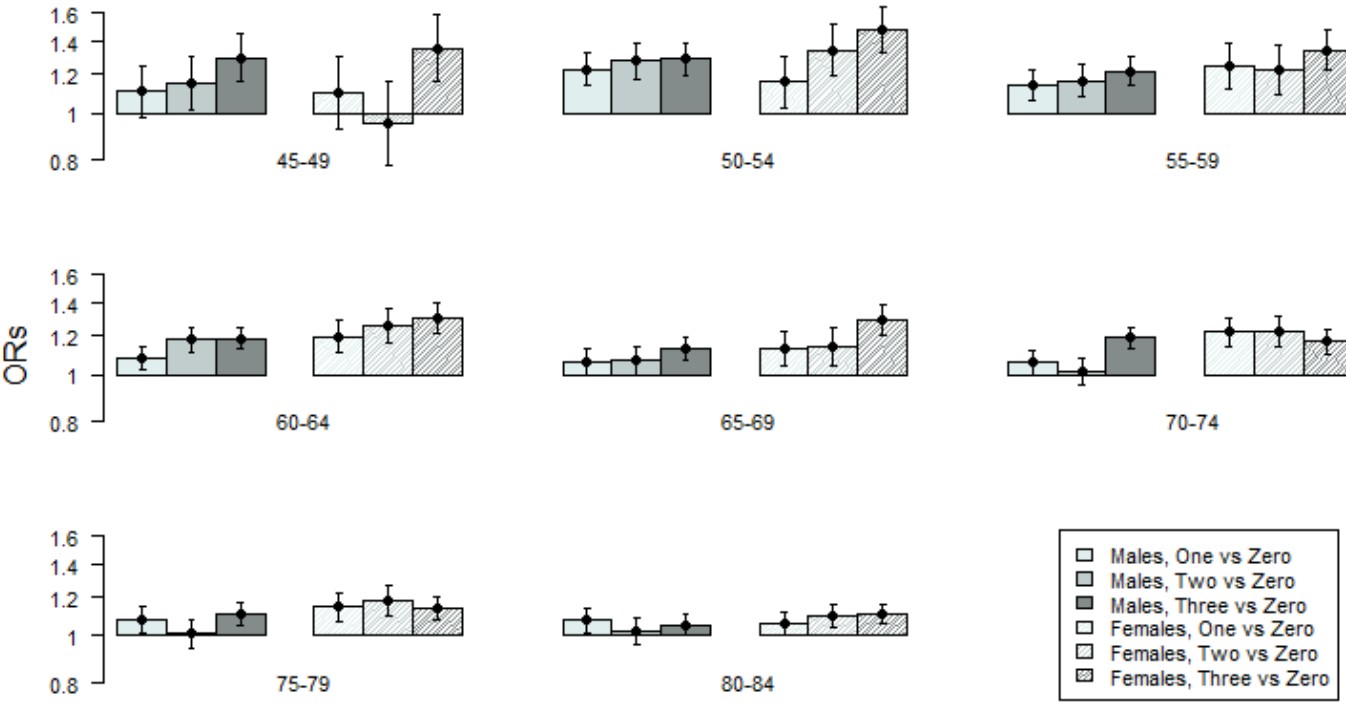

**Figure 2** Adjusted ORs and 95% CIs (on a logarithmic scale) representing the association between various categories of accumulated exposure to neighbourhood deprivation and coronary heart disease using the categorical accumulated model (model 2) in different age cohorts.

exposure over time (table 3). The three sensitivity analyses showed consistent results; the accumulated effect was less pronounced at older ages.

## DISCUSSION

In this study, men and women with the longest accumulated exposure to neighbourhood deprivation had the highest odds of CHD (figure 2), with exception for the oldest age cohorts. The increased neighbourhood association related to an accumulated exposure could be explained by different scenarios. One scenario is that the odds of CHD are consistently increasing with the number of exposed time periods, indicating that the effect of neighbourhood deprivation is monotonously increasing with the time a person resides in such a neighbourhood. If there instead is a tipping point, a further increase in exposure would not result in an additional increasing odds of CHD after a certain level. The main advantage with the statistical models used in the present study was their potential to capture both these scenarios. In men up to 69 years, the odds of CHD consistently increased with the number of periods the men had lived in a deprived neighbourhood. This increase could potentially be described by a continuous variable in men as the AIC value (table 3) for model S2a was lower than that of model 2 in three out of the four youngest cohorts (see online supplementary table 1 for a detailed description of

all models). Such a trend, that is, a continuous increase in odds of CHD by the number of exposed time periods, was not found in women. However, the lower number of CHD events in women, especially in the younger age cohorts, implies that the results are less robust in women than in men. Also, for men and women from 70 years of age and above, we confirmed the previously shown association between residing in a deprived neighbourhood and CHD in all models. However, there was no sign of an increased association with an accumulated exposure to neighbourhood deprivation. In other words, an accumulated effect between exposure to neighbourhood deprivation and CHD was only evident in the younger age cohorts. In addition, in the younger cohorts, the AIC value for model 2 was consistently lower than that of model S2b, which suggests that our accumulation assumption is valid. The sensitivity analyses (models S2a, S2b and S2c) confirmed the main results showing a stronger effect in the younger cohorts and a weaker in the older ones.

That an accumulated exposure of neighbourhood deprivation is associated with increased odds of CHD in the younger but not the oldest age cohorts of men and women suggests that sensitivity to environmental factors involved in the development of CHD may vary with age. The age at exposure could thus be of importance if the sensitivity to the neighbourhood environment is stronger early in life. If this explanation is sufficient, it could be

**Table 3** AIC values (lower is better) from the logistic regression analyses

| | Model 1 Single-point-in-time model | Model 2 Categorical accumulated model | Model S2a Continuous accumulated model, 5-year | Model S2b Timing/period model | Model S2c Continuous accumulated model, 1year |
|---|---|---|---|---|---|
| **Men** | | | | | |
| CHD 45–49 | 25 312.7398 | **25 311.0253** | 25 307.4565 | 25 314.4190 | 25 307.1331 |
| CHD 50–54 | 52 048.9794 | **52 032.2160** | 52 039.0780 | 52 050.9515 | 52 039.0780 |
| CHD 55–59 | 65 542.5041 | **65 534.5155** | 65 533.5186 | 65 535.2344 | 65 540.7682 |
| CHD 60–64 | 109 450.3708 | **109 428.5881** | 109 427.2313 | 109 428.9021 | 109 437.8983 |
| CHD 65–69 | **83 227.1670** | 83 235.3757 | 83 231.8676 | 83 238.7180 | 83 231.1111 |
| CHD 70–74 | 89 818.2465 | **89 814.8284** | 89 820.4081 | 89 814.0974 | 89 818.6223 |
| CHD 75–79 | **73 602.6644** | 73 611.9161 | 73 613.1494 | 73 614.3535 | 73 608.3986 |
| CHD 80–84 | **75 344.0899** | 75 349.1122 | 75 348.7964 | 75 355.5722 | 75 346.9813 |
| **Women** | | | | | |
| CHD 45–49 | **13 587.9857** | 13 592.5468 | 13 595.2763 | 13 596.9268 | 13 593.2401 |
| CHD 50–54 | 27 992.9470 | **27 970.4615** | 27 966.7667 | 27 973.3480 | 27 972.5662 |
| CHD 55–59 | 34 277.0790 | **34 274.6482** | 34 275.9162 | 34 276.5675 | 34 284.0939 |
| CHD 60–64 | 62 174.8598 | **62 160.3900** | 62 162.7564 | 62 166.2798 | 62 176.2077 |
| CHD 65–69 | **55 316.9682** | 55 321.4237 | 55 319.1180 | 55 321.1026 | 55 335.4706 |
| CHD 70–74 | 73 003.5487 | **72 968.0844** | 72 988.6098 | 72 964.1887 | 72 995.1727 |
| CHD 75–79 | 74 455.6142 | **74 440.9304** | 74 447.9954 | 74 445.4397 | 74 453.5558 |
| CHD 80–84 | **96 295.1680** | 96 303.4624 | 96 300.8734 | 96 301.8523 | 96 306.0740 |

The lowest value of models 1 and 2 for each age cohort is in bold. Model 1 represents the single-point-in-time model, and model 2 the categorical accumulated model. Models S2a, S2b and S2c are sensitivity analyses.
AIC, Akaike information criterion; CHD, coronary heart disease.

expected that earlier periods of exposure would have greater impact on the development of CHD than later, that is, in the older cohorts. The results from our sensitivity analysis (model S2b) did not support this hypothesis as exposure during earlier periods did not necessary result in higher ORs (online supplementary table 3 a,b). Survivor bias may also have contributed to weaker associations between neighbourhood deprivation and CHD in older cohorts. Because we studied new-onset CHD, men and women with prior CHD were excluded, and therefore persons who are more sensitive to neighbourhood environmental effects on CHD are more likely to be excluded from older age cohorts.

It is also noteworthy that although the longitudinal assessments of neighbourhood deprivation were of potential importance to assess in the younger age cohorts, they did not considerably improve the prediction of CHD in the population, that is, the AICs were of similar magnitude within each age stratum (table 3). Using a single-point-in-time assessment of neighbourhood deprivation (ie, at baseline) therefore appears to be a reasonable approximation of the exposure to neighbourhood deprivation over time, even during a period as long as 15 years, especially in older age cohorts. The collection of longitudinal assessments, which can be both time-consuming and expensive, is therefore unlikely to have a large impact

on risk prediction, at least among older adults. This is largely a result of the high correlations between the three different 5-year exposure periods (table 2). That these correlations increased with higher age could be a result of that older individuals were less likely to move, or if they move they would move to similar types of neighbourhoods. Mobility has previously been shown to be related to age and family situation.[24 25] Even though a single-point-in-time assessment of neighbourhood deprivation may be equally useful in older age groups, the association between neighbourhood deprivation and CHD was weaker in the older age cohorts, suggesting that other factors than neighbourhood characteristics, as the high age itself, might have the largest influence on CHD. When we used a fixed neighbourhood deprivation measure so that neighbourhood ranking could not change over time, a worse model fit was obtained, although the overall interpretation remained. This also suggests that changes in individuals' deprivation score over time were not driven by changes in deprivation score in their neighbourhoods but rather from the individuals' own mobility.

In the interpretation of the findings of the present study, it is important to keep in mind the conceptual difference between absolute and relative poverty, where absolute poverty implies deprivation of the most basic needs, such as food and shelter, which rarely occurs in

Sweden anymore. However, the negative health effects of relative deprivation are well established, and the social gradient in health by relative deprivation and poverty has been thoroughly described by Sir Michael Marmot in the book 'Status Syndrome'.[26]

There are several limitations to the present study. Negative effects of exposure to neighbourhood deprivation could accumulate over a longer period, and we only had neighbourhood exposure data for a 15-year period. For example, it is possible that individuals' neighbourhood of residence in the ages 20–30 could have had an impact on our results as this is a period in life where most variability in the neighbourhood exposure occurs. We were not able to account for the childhood socioeconomic environment either. However, a Swedish study that examined the association between neighbourhood deprivation and CHD within sibling pairs showed that the association between neighbourhood deprivation and CHD in middle-aged adults was not confounded by genetics or the childhood environment, although slightly confounded in older age groups.[27] These findings suggest that information about neighbourhood deprivation during childhood does not seem to provide any additional information if the neighbourhood exposure in adulthood has been assessed. A possible limitation is that we were only able to follow the individuals for 5 years after the 15 years of exposure. However, the relatively short follow-up period also means that our estimates are unlikely to be overestimated.

A potential limitation of most previous studies is that they are only based on one single assessment of the neighbourhood socioeconomic environment, that is, at baseline. This represents a potential bias because neighbourhoods may change over time and people can move away, which leads to less accurate assessments of the neighbourhood exposure over time. Longitudinal assessments used to create cumulative measures, which was done in the present study, can partly remedy this problem as they take into account possible neighbourhood change and individual mobility over time. Despite this being a strength of the present study, excluding neighbourhood change and mobility could potentially have biased the results in previous studies, although incorporating these factors into a dynamic model as well as how mobility and neighbourhood characteristics interact over time is a challenge.[28] We also checked the mobility in the study population and found that those who had moved during the study period often tended to live in similar types of neighbourhoods over time. Another limitation is that we did not have information on several neighbourhood characteristics that could have health-damaging or health-promoting effects on residents' health. For example, a recent study from the USA reported an association between a healthy food environment and weight loss,[29] which in turn may have a beneficial effect on CHD risk. Furthermore, low social capital is more common in deprived neighbourhoods and is more often associated with poorer access to a regular doctor,[30 31] which is an indirect measure of access to healthcare.[32] Finally, we did not have access to individual lifestyle factors, which may represent important confounders; a previous Swedish study has shown that residents in the most deprived neighbourhoods are at increased risk of being smokers, not performing any physical activity or being obese.[33]

In conclusion, to analyse longitudinal exposure to neighbourhood deprivation is necessary to achieve a deeper understanding of the association between neighbourhood deprivation and CHD. Our results suggest that measures of accumulated exposure may be of greater importance in younger age cohorts and that a hypothesised causality in the association between neighbourhood deprivation and CHD may be possible in younger but not in older age cohorts. Nevertheless, if the focus is solely on prediction, a model based on single-point-in-time assessments may be an adequate approximation, at least in older age cohorts.

**Contributors** KS, OM and CC were responsible for the initiation and conception of the study. SLL and KS designed the study and drafted the manuscript. SLL performed the statistical analysis. All authors have contributed to the interpretation of the results and took part in finalising the manuscript The final manuscript has been approved by all the authors, and all four can take public responsibility for the content of the manuscript.

**Funding** This work was supported by grants from the Swedish Research Council to KS, the Swedish Heart-Lung Foundation, and the National Heart, Lung, and Blood Institute of the National Institutes of Health under Award Number R01HL116381 to KS.

**Competing interests** None declared.

**Patient consent for publication** Not required.

**Ethics approval** The study was approved by the Regional Ethics Committee in Lund, Sweden (Dnr 2012/795).

**Provenance and peer review** Not commissioned; externally peer reviewed.

**Data availability statement** No data are available.

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
