## [Reviewer comments · BMJ Open]

ARTICLE DETAILS

TITLE (PROVISIONAL)	Accumulated neighborhood deprivation and Coronary Heart Disease: A nationwide cohort study from Sweden
AUTHORS	Lönn, Sara; Melander, Olle; Crump, Casey; Sundquist, Kristina

VERSION 1 – REVIEW

REVIEWER	Emily T Murray University College London United Kingdom
REVIEW RETURNED	18-Feb-2019

GENERAL COMMENTS	In this work the authors used Swedish nationwide data of men and women ages 45 and above to examine if an accumulated exposure to neighbourhood deprivation resulted in higher risks of CHD. In general, I think this is a well-written paper that brings the field of neighbourhood effects on cardiovascular outcomes forward. However, I think there are a few issues that need to be addressed. My main concern is the collapsing the five-year neighbourhood deprivation scores into binary exposed to a deprived neighbourhood during the 5 years or not. First, the definition of exposure to a 'deprived' neighbourhood is unclear. Is this a relative score above 1? Second, and more concerning though, is that the dichotomisation loses a lot of the exposure information and potentially creates misclassification in the accumulation score. At the minimum, it would be good to show in sensitivity analysis how the individual five-year deprivation scores are associated with CHD (linear?). But also a continuous accumulation score association with CHD. The second is that I think there needs to be more discussion about changed in neighbourhoods over time & relative vs absolute neighbourhood deprivation. I find it interesting that for the younger cohort it is the two periods of exposure that have the highest CHD incidence. It may be worth thinking about what was happening historically in each of the 5-year periods. Right now the deprivation measure is relative, so that if the entire country became more deprived during a period, that person's relative deprivation standing would stay the same if their neighbourhood became worse as well. But if their neighbourhood didn't become as worse as others, could it now be seen as relatively 'unexposed'? Other points for clarification: 1. Abstract, last sentence & discussion last sentence. You have only looked at correlations of neighbourhood deprivation in people aged 30 at the youngest. Most variability in neighbourhood deprivation occurs when people leave parental homes and
--

	throughout their 20's-30's, so I think single point in time assessments may only be a good predictor in older adults. 2. Methods, page 5, line 5: I found it hard to piece together when exposure and outcome periods/ages occurred. Maybe a figure would help? 3. Methods, page 5, line 10: what did you link to? 4. Measures, page, line 45: are the four dimensions of deprivation defined somewhere? 5. Measures, page 6, line 27: how can you assume those missing education are in the low category? I would think more appropriate to make a missing category, exclude them from the analysis or impute. 6. Results, page 8, lines 8-9: how did you create the low, medium or high deprived neighbourhood categories? This isn't in the methods. 7. Results, page 8, lines 51-52: accumulation model a bit fit until age 64, but then what? In general, seems AIC values aren't that different for most age groups. Would you say that's true? 8. Discussion, page 9, lines 45-50: I also wonder what the death rates by deprivation were over the 5-year follow-up period? Or were they excluded because they needed to have CHD data? 9. Discussion, page 10, lines 23-24: there are references from the residential mobility literature that show that older individuals are less like to move, and subsequently change deprivation levels, than younger individuals. 10. Tables 1a and 1b. Hard to read across the rows. Maybe make every other age group have a light grey background? 11. Table 3. Please define model 1 and model 2. Which is single point in time and which accumulation? 12. Think there are some missing references: a. Carson AP 2007 – Annals of Epi 17(4):296-303. b. Lemelin ET 2009 – SSM 68(3):444-451. c. Nordstrom CK 2004-SSM December 2004.
--	---

REVIEWER	Tim Morris University of Bristol, UK.
REVIEW RETURNED	19-Feb-2019

GENERAL COMMENTS	The authors present a well written manuscript investigating neighbourhood deprivation accumulated over 15 years and coronary heart disease during a five-year period following neighbourhood measurement. The data they use are fantastic, coming from a large sample of Swedish nationwide data. As such this study can make an important contribution to the literature. In my opinion the manuscript as it is currently presented is let down by a strange choice of Tables, lack of clarity on the results, and a lack of awareness of the study limitations. More careful thought on the presentation of results, a clearer explanation of methods and a more thorough discussion would make the paper far stronger. My specific comments are summarised below, I hope that these prove helpful to the authors and I look forward to seeing a revised version of the manuscript. The approach of investigating dose response associations is appropriate to the research question. However, dose response relationships should be considered as consistent with causal relationships rather than indicating them. I realise that this may seem a little pedantic, but I feel that care should be taken with causal terminology given that the authors do not use a method for causal analysis. This is particularly true given the tricky issue of
---

confounding; is the neighbourhood deprivation dose response totally independent of all other confounding factors? I very much doubt it. The neighbourhood measures are aggregates and so reflect their constituent individuals, meaning that they could play no causal role even where dose response associations are observed. This isn't a problem with the analysis, but I think warrants a toning down of causal terminology in places.

The authors only use neighbourhood exposure data for 15 years and follow-up data for 5 years. I think that this should be expanded upon in the limitations as it seems reasonable to me that the negative effects of exposure to neighbourhood deprivation could accumulate over a longer period.

It sounds to me that the same neighbourhood can have different values at different times because of the way the data were coded. Is this true, and if so, how many neighbourhoods change deprivation category over time? If you use a fixed deprivation measure so that neighbourhoods cannot change ranking, are your results consistent?

In your statistical analyses it isn't clear how you create low, middle and high neighbourhood deprivation for the single point in time measure. I assume that this reflects "highly deprived" (>1 SD), "affluent" (<1 SD), and all other neighbourhoods? This is central to the paper and needs to be made explicitly clear.

In table 1a and table 1b, it really confused me that you presented results for each of the individual exposure periods rather than the combined exposure periods which you go on to use in your analyses. It doesn't come across from your manuscript that you are specifically interested in the timing of deprivation, but rather the duration of exposure, yet the appearance of this table makes it seem that you are interested in the timing. Combining the categories would make the tables not only more digestible but also more focussed on your research question. Your main analyses for Model 2 and Figure 2 uses the combined duration categories which makes the inclusion of the non-combined categories in T1a & T1b all the more confusing.

I can't see anywhere a Table that displays the data underlying Figure 2. This needs to be included and it is absolutely imperative that you include information in the Figure legends or footnotes explaining which Table the data come from. EDIT: I now wonder if the data in Figure 2 is from the left-hand models of supplementary Tables T2a and T2b? In which case you need a far clearer justification of why you chose only these categories to model. Personally, I think that you should combine the categories and report these associations instead. When you account for all of the categories instead of just the 3 that you have selected, the associations which you highlight in the left-hand model change considerably and no longer provide the largest effect size estimates. This means that your statement in the results that "Between ages 45 years and 79 years in men and between ages 45 years and 69 years in women, those in the three time-periods' exposure category had the strongest associations with CHD" is no longer true. Effectively, your sensitivity analyses provide a better fit to the data than your main analysis. This likely explains why your estimates change so much.

	In Table 2 it isn't immediately clear what you are correlating. A footnote here would be really useful to clarify this to the reader. I'm also unsure what Table 2 adds to your manuscript; it is unsurprising that correlations are higher where the periods are temporally closer than further away, and again it confuses the reader about whether you are looking at timing or duration of exposure to neighbourhood deprivation. That older people move less than younger people and that people are more likely to move the longer they remain in a place is entirely unsurprising and supported by a huge body of demographic research. Maybe this would be better in the supplementary material? I don't really understand why Table 3 is included in the main manuscript as you do not reference it anywhere. In supplementary Tables T2a and T2b you need titles and footnotes to clearly explain what the tables show and that you only examine some associations (0,1,0; 1,0,1; 1,1,1) in the left-hand side model (I assume that this is Model 1 but it isn't labelled). You also need to state your justification for doing this. As above, I don't think that this is particularly relevant to your research question and as such it confuses the reader more than it clarifies. You need to devote considerable space to your discussion to the limitations of your study. The fundamental limitation in my opinion is that there is no discussion of confounding factors and how these could bias your results. As I have highlighted above, the fact that neighbourhoods are a product of their individuals means that your results are very suspect to individual level confounding factors. You use income and education but there are numerous additional confounding factors in the association between neighbourhood deprivation and CHD that you are unable to account for and as such there will be considerable unobserved confounding in your study. It surprised me that there was no discussion of residential or neighbourhood mobility/migration. There is a growing body of evidence that neighbourhood effects may more strongly represent the impact of changing residential/neighbourhood environment than of the environment per se. I assume that the majority of neighbourhood deprivation changes in your dataset are the result of moving instead of the result of neighbourhoods changing over time (though as I mention above this isn't clear from your methods section), so this issue should be at least briefly discussed.
--	---

VERSION 1 – AUTHOR RESPONSE

Reviewer(s)' Comments to Author:

Reviewer: 1

Reviewer Name: Emily T Murray

Institution and Country: University College London

United Kingdom

Please state any competing interests or state 'None declared': None declared.

#1. In this work the authors used Swedish nationwide data of men and women ages 45 and above to examine if an accumulated exposure to neighbourhood deprivation resulted in higher risks of CHD. In general, I think

this is a well-written paper that brings the field of neighbourhood effects on cardiovascular outcomes forward. However, I think there are a few issues that need to be addressed. My main concern is the collapsing the five-year neighbourhood deprivation scores into binary exposed to a deprived neighbourhood during the 5 years or not.

We appreciate the overall positive comments on our manuscript. We also acknowledge the Reviewer's concerns over defining our exposure variable into the binary variable exposed to a deprived neighbourhood or not. When we designed our study and defined the exposure variable, we considered using several different approaches. These approaches included dividing the entire time period into shorter time intervals as well as modelling the underlying continuous variable. However, most of these approaches would have implied difficulties in the description and interpretation of the results, especially because our study is the first to explore the association between an accumulated measure of neighbourhood deprivation and coronary heart disease (CHD). In addition, as most previous studies of the association between neighbourhood deprivation and CHD have used a categorical neighbourhood variable, we wanted to be consistent with previous studies and therefore chose to use a categorical variable instead of a continuous one. This approach also makes the results of the present study comparable with previous research that were not able to include an accumulated measure. However, we have performed a sensitivity analysis based on a continuous neighbourhood variable and the results from the sensitivity analysis confirmed our main results. Please also see our response to comment 3 below.

#2. First, the definition of exposure to a 'deprived' neighbourhood is unclear. Is this a relative score above 1?

Yes, it is a relative score. This is described in the sentence: "A highly deprived neighborhood was defined as a neighborhood with a deprivation score over 1, and an affluent neighborhood was defined as a neighborhood with a deprivation score under -1".

#3. Second, and more concerning though, is that the dichotomisation loses a lot of the exposure information and potentially creates misclassification in the accumulation score. At the minimum, it would be good to show in sensitivity analysis how the individual five-year deprivation scores are associated with CHD (linear?). But also a continuous accumulation score association with CHD.

The overall aim of this study was to evaluate if there is an accumulated effect of exposure to neighbourhood deprivation on the risk of CHD. To construct a model that could be useful for this purpose, one necessary condition is that the model cannot assume what we intend to investigate (Kriebel, Checkoway et al. 2007, de Vocht, Burstyn et al. 2015). In other words, it is not possible to assume an increasing association by the number of years a person resides in a deprived neighbourhood and interpret an OR over 1 as a result of an accumulated effect. In addition, our goal was to use a relatively simple model that would be easy to understand for the broad audience of the BMJ Open where some readers may have little statistical training.

As mentioned above (comment 1), we considered dividing the entire time period into shorter time intervals. However, such an approach would have increased the number of variables, which is problematic especially for the younger cohort where there are fewer CHD cases.

However, as we were able to provide evidence of an accumulated effect in the younger cohorts, the next step is to explore whether the associations are linear or not. Two new sensitivity analysis were therefore added; one where the number of five-year-periods is treated as a linear term and one based on the number of years a person resides in a deprived neighbourhood. The following sentence is new (please see the last sentence in the first paragraph in the discussion section):

"The sensitivity analyses (Model S2a, S2b and S2c) confirmed the main results showing a stronger effect in the younger cohorts and a weaker in the older ones."

#4. The second is that I think there needs to be more discussion about changed in neighbourhoods over time & relative vs absolute neighbourhood deprivation. I find it interesting that for the younger cohort it is the two periods of exposure that have the highest CHD incidence. It may be worth thinking about what was happening historically in each of the 5-year periods. Right now the deprivation measure is relative, so that if the entire country became more deprived during a period, that person's relative deprivation standing would stay the

same if their neighbourhood became worse as well. But if their neighbourhood didn't become as worse as others, could it now be seen as relatively 'unexposed'?

We appreciate this relevant comment on neighbourhood change over time. Most previous follow-up studies were only based on one single assessment of the neighbourhood socioeconomic environment, i.e., at baseline. This represents a methodological limitation of previous studies because neighbourhoods may change over time and people can move away, which leads to less representative assessments of the neighbourhood exposure over time. Longitudinal assessments can be used to create cumulative measures, which can partly remedy this problem as they take into account possible neighbourhood change and individual mobility over time. We have added this as a potential strength in the discussion section.

We have also added a discussion on relative vs. absolute deprivation. The following text on relative vs. absolute deprivation is new:

"It is also important to keep in mind the conceptual difference between absolute and relative poverty where absolute poverty implies deprivation of the most basic needs, such as food and shelter, which rarely occurs in Sweden anymore. However, the negative health effects of relative deprivation are well established and the social gradient in health by relative deprivation and poverty has been thoroughly described by Sir Michael Marmot in the book "Status Syndrome"."

Sweden experienced only one major financial crisis during the exposure period, i.e., in the beginning of the 90s, when the GNP dropped between 1.3% and 2.0% during three consecutive years. As a result, house prices dropped with around 20% and unemployment rates rose. During a short period of time in 1992, mortgage rates rocketed up to 24%. We could, however, not observe that the rates of CHD were substantially higher during this time period.

#5. Other points for clarification:

#5.1. Abstract, last sentence & discussion last sentence. You have only looked at correlations of neighbourhood deprivation in people aged 30 at the youngest. Most variability in neighbourhood deprivation occurs when people leave parental homes and throughout their 20's-30's, so I think single point in time assessments may only be a good predictor in older adults.

This is a valuable comment; unfortunately, we do not have information from the phase in life where most people leave their parental home. We have mentioned this as a potential limitation.

#5.2. Methods, page 5, line 5: I found it hard to piece together when exposure and outcome periods/ages occurred. Maybe a figure would help?

We apologize for being unclear and have rephrased the parts in the methods section describing when exposure and outcome occurred. Please see the first paragraph in the methods section. Due to lack of space, we prefer not to add a new figure and hope that the revision of the methods section has helped to improve the clarity.

#5.3. Methods, page 5, line 10: what did you link to?

We linked several nationwide Swedish registers and geographical information using a unique personal identification number, replaced by a pseudonymized serial number. These registers are now mentioned after the sentence including "linked".

#5.4. Measures, page, line 45: are the four dimensions of deprivation defined somewhere?

We have mentioned the four dimensions of deprivation, i.e., low income, low education, unemployment, and receipt of social welfare.

#5.5. Measures, page 6, line 27: how can you assume those missing education are in the low category? I would think more appropriate to make a missing category, exclude them from the analysis or impute.

Among Swedish-born residents (representing 90% of the study population) who fulfil our inclusion criteria, less than 0.1% have missing information on education; the corresponding figure for foreign-born residents is less than 0.5%. A separate category of those with missing education is therefore not meaningful. The reason for categorizing missing education as low is because missing education is among Swedish-born in most cases associated with no high-school or university degree. Foreign-born residents with missing education means that there is no registered education in Sweden, no documents of a degree or that the immigrant did not report his or her level of education to the authorities. We have clarified this in the text and mentioned the proportion of those with missing education.

#5.6. Results, page 8, lines 8-9: how did you create the low, medium or high deprived neighbourhood categories? This isn't in the methods.

This text has now been revised. Please see the second paragraph in the text with the subheading "measures":

"The aggregated measure was standardized to have mean 0 and standard deviation 1 each year, making it a relative measure comparable between years. A highly deprived neighbourhood was defined as a neighbourhood with a deprivation score over 1, and an affluent neighbourhood was defined as a neighbourhood with a deprivation score under -1."

#5.7. Results, page 8, lines 51-52: accumulation model a bit fit until age 64, but then what? In general, seems AIC values aren't that different for most age groups. Would you say that's true?

This is correct and has now been clarified.

#5.8. Discussion, page 9, lines 45-50: I also wonder what the death rates by deprivation were over the 5-year follow-up period? Or were they excluded because they needed to have CHD data?

Death rates were included and this has now been clarified in the text and tables.

#5.9. Discussion, page 10, lines 23-24: there are references from the residential mobility literature that show that older individuals are less like to move, and subsequently change deprivation levels, than younger individuals.

We have now added two reference that examines residential mobility in older individuals.

#5.10. Tables 1a and 1b. Hard to read across the rows. Maybe make every other age group have a light grey background?

Thanks for pointing this out. We have revised Tables 1a and 1b for clarity. For example, we added a horizontal line between each age cohort and combined the exposure groups.

#5.11. Table 3. Please define model 1 and model 2. Which is single point in time and which accumulation?

This has now been clarified.

#5.12. Think there are some missing references:

- a. Carson AP 2007 – Annals of Epi 17(4):296-303.
- b. Lemelin ET 2009 – SSM 68(3):444-451.
- c. Nordstrom CK 2004-SSM December 2004.

We have added these missing references to the reference list. Please see the introduction.

Reviewer: 2

Reviewer Name: Tim Morris

Institution and Country: University of Bristol, UK.

Please state any competing interests or state 'None declared': None declared
Please leave your comments for the authors below

#1. The authors present a well written manuscript investigating neighbourhood deprivation accumulated over 15 years and coronary heart disease during a five-year period following neighbourhood measurement. The data they use are fantastic, coming from a large sample of Swedish nationwide data. As such this study can make an important contribution to the literature. In my opinion the manuscript as it is currently presented is let down by a strange choice of Tables, lack of clarity on the results, and a lack of awareness of the study limitations. More careful thought on the presentation of results, a clearer explanation of methods and a more thorough discussion would make the paper far stronger. My specific comments are summarised below, I hope that these prove helpful to the authors and I look forward to seeing a revised version of the manuscript.

We appreciate the positive overall comments about the study and the very useful comments, which we believe have helped us to improve the manuscript.

#2. The approach of investigating dose response associations is appropriate to the research question. However, dose response relationships should be considered as consistent with causal relationships rather than indicating them. I realise that this may seem a little pedantic, but I feel that care should be taken with causal terminology given that the authors do not use a method for causal analysis. This is particularly true given the tricky issue of confounding; is the neighbourhood deprivation dose response totally independent of all other confounding factors? I very much doubt it. The neighbourhood measures are aggregates and so reflect their constituent individuals, meaning that they could play no causal role even where dose response associations are observed. This isn't a problem with the analysis, but I think warrants a toning down of causal terminology in places.

As the reviewer points out, a causal relationship in general leads to a dose-response association but an estimated dose-response association is not necessary a result of a causal relationship. Residual confounding is rather likely to be present in an observational study such as ours. We have toned down the causal terminology throughout the manuscript. Please also see the revised first paragraph in the introduction.

#3. The authors only use neighbourhood exposure data for 15 years and follow-up data for 5 years. I think that this should be expanded upon in the limitations as it seems reasonable to me that the negative effects of exposure to neighbourhood deprivation could accumulate over a longer period.

This is an important comment that we have incorporated as a potential limitation in the discussion section.

#4. It sounds to me that the same neighbourhood can have different values at different times because of the way the data were coded. Is this true, and if so, how many neighbourhoods change deprivation category over time? If you use a fixed deprivation measure so that neighbourhoods cannot change ranking, are your results consistent?

Most previous follow-up studies were only based on one single assessment of the neighbourhood socioeconomic environment, i.e., at baseline. This represents a methodological limitation of previous studies because neighbourhoods may change over time and people can move away, which leads to less accurate assessments of the neighbourhood exposure over time. Longitudinal assessments can be used to create cumulative measures, which can partly remedy this problem as they take into account possible neighbourhood change and individual mobility over time. We have added this as a potential strength in the discussion section.

We also used a fixed deprivation measure so that neighbourhoods could not change ranking over time. We used a deprivation score from one point in time (year 2000) and assumed that all neighbourhoods had the same deprivation score during the whole time period. This provided a worse model fit but the overall interpretation could not be rejected.

#5. In your statistical analyses it isn't clear how you create low, middle and high neighbourhood deprivation for the single point in time measure. I assume that this reflects "highly deprived" (>1 SD), "affluent" (<1 SD), and all other neighbourhoods? This is central to the paper and needs to be made explicitly clear.

Thanks for pointing this out. We have now clarified this in the methods section.

“The aggregated measure was standardized to have mean 0 and standard deviation 1 each year, making it a relative measure comparable between years. A highly deprived neighbourhood was defined as a neighbourhood with a deprivation score over 1, and an affluent neighbourhood was defined as a neighbourhood with a deprivation score under -1.”

#6. In table 1a and table 1b, it really confused me that you presented results for each of the individual exposure periods rather than the combined exposure periods which you go on to use in your analyses. It doesn't come across from your manuscript that you are specifically interested in the timing of deprivation, but rather the duration of exposure, yet the appearance of this table makes it seem that you are interested in the timing. Combining the categories would make the tables not only more digestible but also more focussed on your research question. Your main analyses for Model 2 and Figure 2 uses the combined duration categories which makes the inclusion of the non-combined categories in T1a & T1b all the more confusing.

We have now modified Table 1a and 1b to reflect the combined exposure variables we focus on in the main analysis. The categories that reflect the timing of the exposure are only used in a sensitivity analysis.

#7. I can't see anywhere a Table that displays the data underlying Figure 2. This needs to be included and it is absolutely imperative that you include information in the Figure legends or footnotes explaining which Table the data come from. EDIT: I now wonder if the data in Figure 2 is from the left-hand models of supplementary Tables T2a and T2b? In which case you need a far clearer justification of why you chose only these categories to model. Personally, I think that you should combine the categories and report these associations instead. When you account for all of the categories instead of just the 3 that you have selected, the associations which you highlight in the left-hand model change considerably and no longer provide the largest effect size estimates. This means that your statement in the results that “Between ages 45 years and 79 years in men and between ages 45 years and 69 years in women, those in the three time-periods' exposure category had the strongest associations with CHD” is no longer true. Effectively, your sensitivity analyses provide a better fit to the data than your main analysis. This likely explains why your estimates change so much.

This is correct and the data from Figure 2 were supposed to be presented in Tables T2a and T2b. However, we are embarrassed to say that we had accidentally included a previous version of the tables that was not correct. Thank you for noticing! We apologize for this mistake and the confusion this has caused during the review. In the updated and corrected version, Model S2b does no longer provide a better fit to the data.

#8. In Table 2 it isn't immediately clear what you are correlating. A footnote here would be really useful to clarify this to the reader. I'm also unsure what Table 2 adds to your manuscript; it is unsurprising that correlations are higher where the periods are temporally closer than further away, and again it confuses the reader about whether you are looking at timing or duration of exposure to neighbourhood deprivation. That older people move less than younger people and that people are more likely to move the longer they remain in a place is entirely unsurprising and supported by a huge body of demographic research. Maybe this would be better in the supplementary material?

I don't really understand why Table 3 is included in the main manuscript as you do not reference it anywhere.

We have clarified the table headings in Table 2. The intention with this table is to describe the longitudinal exposure measure in more detail. We believe that the content in Table 2 illustrates why an accumulated exposure is less predictive at older ages and has clarified this in the revised manuscript. Therefore we would like to keep this table in the main manuscript but could move it to the supplementary material if this still is requested.

For Table 3, we apologize for not referring to the table in the manuscript. We have now added information that we refer to Table 3 in the results and discussion section.

#9. In supplementary Tables T2a and T2b you need titles and footnotes to clearly explain what the tables show and that you only examine some associations (0,1,0; 1,0,1; 1,1,1) in the left-hand side model (I assume that this is Model 1 but it isn't labelled). You also need to state your justification for doing this. As above, I don't

think that this is particularly relevant to your research question and as such it confuses the reader more than it clarifies.

We apologize for being unclear. We have now added information in the tables on how the models are designated (e.g., Model 2, Model 2a etc.). The justification for doing this is now explained at the end of the introduction. The following text is new:

“When investigating the potential existence of an accumulated “effect” it is, however, not possible to a priori decide which metric that it most suitable for the analysis; instead, it is necessary to analyse various measures and compare how well the models fit the data. (Kriebel, Checkoway et al. 2007, de Vocht, Burstyn et al. 2015). One crucial condition is therefore not to assume a dose-response relationship in the model specification or, in other words, to assume a linear increase in the associations.”

#10. You need to devote considerable space to your discussion to the limitations of your study. The fundamental limitation in my opinion is that there is no discussion of confounding factors and how these could bias your results. As I have highlighted above, the fact that neighbourhoods are a product of their individuals means that your results are very suspect to individual level confounding factors. You use income and education but there are numerous additional confounding factors in the association between neighbourhood deprivation and CHD that you are unable to account for and as such there will be considerable unobserved confounding in your study. It surprised me that there was no discussion of residential or neighbourhood mobility/migration. There is a growing body of evidence that neighbourhood effects may more strongly represent the impact of changing residential/neighbourhood environment than of the environment per se. I assume that the majority of neighbourhood deprivation changes in your dataset are the result of moving instead of the result of neighbourhoods changing over time (though as I mention above this isn't clear from your methods section), so this issue should be at least briefly discussed.

We have added a discussion about possible confounders and reflected upon how these could bias our results. The limitations section also includes a discussion on neighborhood change and mobility. The use of an accumulated measure should, however, partly remedy this limitation. We also checked the mobility in the study population and found that those who had moved during the study period often tended to live in similar types of neighbourhoods over time. This information has been added to the limitations section.

References

de Vocht, F., I. Burstyn and N. Sanguanchaiyakrit (2015). "Rethinking cumulative exposure in epidemiology, again." *J Expo Sci Environ Epidemiol* **25**(5): 467-473.
Kriebel, D., H. Checkoway and N. Pearce (2007). "Exposure and dose modelling in occupational epidemiology." *Occupational and environmental medicine* **64**(7): 492-498.

VERSION 2 – REVIEW

REVIEWER	Emily Murray University College London, United Kingdom
REVIEW RETURNED	21-May-2019
GENERAL COMMENTS	Thanks to the author(s) for addressing some of my previous comments. The manuscript has been improved on these points. Now that I have clarity on how the neighbourhood deprivation measure and exposure periods defined, I still find myself confused as to what models were being tested in the paper (see below) and

	whether these models match up with the stated aim of the paper. I believe the paper could still do with a re-write to clarify what life course models were tested (sensitive period, accumulation, trajectory, etc), how each models fits in with the research question and adding descriptors to results tables to link estimates back to the particular life course model being shown. A. Previous #3. The overall aim of this study was to evaluate if there is an accumulated effect of exposure to neighbourhood deprivation on the risk of CHD. To construct a model that could be useful for this purpose, one necessary condition is that the model cannot assume what we intend to investigate (Kriebel, Checkoway et al. 2007, de Vocht, Burstyn et al. 2015). In other words, it is not possible to assume an increasing association by the number of years a person resides in a deprived neighbourhood and interpret an OR over 1 as a result of an accumulated effect. I find the logic in this statement a bit puzzling. If you're research question is whether the effect of neighbourhood deprivation on CHD accumulates over time/life course, then how will you test that without testing a model that assumes just that? From reading your paper, it is my understanding that you have assumed that exposure to neighbourhood deprivation is related to CHD occurs in different ways & then test which model has the best fit. My understanding is that the models are as follows: Model 1: Whether an individual has lived in a deprived neighbourhood at any time in each of the three five-year periods before the baseline. You call this a single-point-in-time measure, but I believe you have created more of a period prevalence measure. I would see single-point-in-time to be what the deprivation was in the neighbourhood they were residing at baseline. Or have I misunderstood the text in the measures section? Model 2: The eight patterns of longitudinal exposure I would see as trajectory models, rather than strict accumulation. Model 2a: I would consider this model more in line with your stated research question. More time periods exposed equals more CHD risk. Model 2b: How is this different than model 2? Model 2c: How does this differ from model 2a? Suggestions:  1. Equations for each model would be extremely helpful for knowing exactly what you are modelling. There are some great papers, such as this one by Mishra et al (https://academic.oup.com/ije/article/38/2/528/655570), where each model to be tested is stated explicitly. At the moment, I can't tell what was fitted for each model by the text alone. 2. If your hypothesis is accumulation, then stick to testing accumulation models. If you're interested in whether the order/timing of when neighbourhood deprivation occurs, then test
--	--

	the mobility hypothesis (model 2 I believe). But if you include the latter, you need to add text to the introduction and discussion on justification for why you think timing of neighbourhood deprivation would matter for CHD risk. 3. It would also help if in the methods section you clearly defined what each model represented. Add these labels Table 3. B. #5.4. Measures, page, line 45: are the four dimensions of deprivation defined somewhere? We have mentioned the four dimensions of deprivation, i.e., low income, low education, unemployment, and receipt of social welfare. By 'definition' I meant how would you know a neighbourhood was classified as 'low income'? Was there a certain average neighbourhood income, certain percentage of households in the neighbourhood below a certain income threshold, etc? C. Supplementary tables: Why are only three estimates listed for models 2(crude), 2(adjusted) and S2a? There needs to be a lot more description on these tables to let the reader be able to understand what is being presented and results.
--	--

REVIEWER	Tim Morris University of Bristol, UK
REVIEW RETURNED	11-Apr-2019

GENERAL COMMENTS	The authors have done a thorough job in revising their manuscript and I congratulate them for their efforts, which have made the paper far stronger. I believe that the manuscript is suitable for publication as it is, but that it would benefit from some additional information as I highlight below. These changes may not be necessary for publication, but they will help readers to interpret the results and appreciate the hard work that the authors have put into this study. 1. I don't fully understand the argument that you are making in the third paragraph of your introduction and your response to reviewer #1. I disagree with the sentence in the introduction starting "one crucial condition...". It is perfectly valid to assume a dose-response relationship (indeed you seem to argue this in the first paragraph of the introduction when you cite Hill's criteria), but this assumption may not hold in the data when you test it as you have done. Personally, I think that removing this sentence would be of benefit to the manuscript. 2. It would be very helpful to the reader to explain which 15 years you are referring to when you say "during 15 years of potential exposure" in the 'Study sample' section of the Methods. This doesn't seem to be clearly explained anywhere. My best guess is that it is the years 1988-1992 as your 5 years of follow-up appears to be 2003-2007, though this is also not explicitly stated. Clarity here will really help others to interpret and contextualise your results correctly. 3. You make clear in your responses to review that deprivation of neighbourhood's doesn't change over time in your dataset, as you
--

	say: "We also used a fixed deprivation measure so that neighbourhoods could not change ranking over time. We used a deprivation score from one point in time (year 2000) and assumed that all neighbourhoods had the same deprivation score during the whole time period. This provided a worse model fit but the overall interpretation could not be rejected.". This information should be included in your methods when you discuss neighbourhood deprivation as it is important to your study. I also think that you should mention this in your discussion as it directly impacts the interpretation of your results, though it will unlikely prove to be a large source of bias (see: Norman P. 2010. Identifying Change Over Time in Small Area Socio-Economic Deprivation. Applied Spatial Analysis and Policy). You argue in your responses that this is a potential strength of your paper, but it is also a potential limitation. 4. You don't include in your discussion that your follow-up period was only 5 years, which may be too short for effects to manifest and be detected. This is important as I would argue that it means you under estimate associations and therefore it strengthens your findings. 5. You don't refer to Model S2c in your results. I find this strange, as I think it provides a lot of strength to your findings. It provides evidence that your results aren't driven by dichotomisation and misclassification problem raised by reviewer #1. 6. Second paragraph of discussion says that the findings "neither supported nor contradicted this hypothesis" which sounds odd. Surely it's one or the other? Minor housekeeping edits that the authors may wish to make:  - Typo in third paragraph of introduction: "decide which metric that it most suitable". - In 'study sample' would confidentiality be a more appropriate word than "integrity"? - Typo in patient and public involvement "advice" should be advise. - Would Table 3 be better suited to the supplementary material?
--	---

VERSION 2 – AUTHOR RESPONSE

Reviewer(s)' Comments to Author:

Reviewer: 2

Reviewer Name: Tim Morris

Institution and Country: University of Bristol, UK Please state any competing interests or state 'None declared':None declared

Please leave your comments for the authors below The authors have done a thorough job in revising their manuscript and I congratulate them for their efforts, which have made the paper far stronger. I believe that the manuscript is suitable for publication as it is, but that it would benefit from some additional information as I highlight below. These changes may not be necessary for publication, but they will help readers to interpret the results and appreciate the hard work that the authors have put into this study.

Response: Thank you for your appreciation of our revised manuscript and comments on how to improve the paper further prior to publication. Please see our detailed responses below.

1. I don't fully understand the argument that you are making in the third paragraph of your introduction and your response to reviewer #1. I disagree with the sentence in the introduction starting "one crucial condition...". It is perfectly valid to assume a dose-response relationship (indeed you seem to argue this in the first paragraph of the introduction when you cite Hill's criteria), but this assumption may not hold in the data when you test it as you have done. Personally, I think that removing this sentence would be of benefit to the manuscript.

Response: We have removed that sentence. In addition, we have added new text (and a new reference by Mishra et al., suggested by Reviewer #1) in order to justify the use of different models for investigating our aims.

2. It would be very helpful to the reader to explain which 15 years you are referring to when you say "during 15 years of potential exposure" in the 'Study sample' section of the Methods. This doesn't seem to be clearly explained anywhere. My best guess is that it is the years 1988-1992 as your 5 years of follow-up appears to be 2003-2007, though this is also not explicitly stated. Clarity here will really help others to interpret and contextualise your results correctly.

Response: The explanation of which 15 years we are referring to in the paragraph with the subheading "Study sample" can be found in the paragraph with the subheading "Measures". We have now added a parenthesis in the paragraph with the subheading "Study sample" that refers to the paragraph with the subheading "Measures": "...during 15 years of potential exposure (see Measures below)." We have also revised the paragraph with the subheading "Measures" in order to further clarify which years that represent the exposure period.

3. You make clear in your responses to review that deprivation of neighbourhood's doesn't change over time in your dataset, as you say: "We also used a fixed deprivation measure so that neighbourhoods could not change ranking over time. We used a deprivation score from one point in time (year 2000) and assumed that all neighbourhoods had the same deprivation score during the whole time period. This provided a worse model fit but the overall interpretation could not be rejected.". This information should be included in your methods when you discuss neighbourhood deprivation as it is important to your study. I also think that you should mention this in your discussion as it directly impacts the interpretation of your results, though it will unlikely prove to be a large source of bias (see: Norman P. 2010. Identifying Change Over Time in Small Area Socio-Economic Deprivation. Applied Spatial Analysis and Policy). You argue in your responses that this is a potential strength of your paper, but it is also a potential limitation.

Response: We have added this information to the methods section (second last sentence, just before the results) as well as in the discussion (third paragraph).

4. You don't include in your discussion that your follow-up period was only 5 years, which may be too short for effects to manifest and be detected. This is important as I would argue that it means you under estimate associations and therefore it strengthens your findings.

Response: Thank you for pointing this out. We have added the following new text to the discussion section: "A potential limitation is that we were only able to follow the individuals for five years after the 15-years' of exposure. However, the relatively short follow-up period also means that our estimates are unlikely to be overestimated."

5. You don't refer to Model S2c in your results. I find this strange, as I think it provides a lot of strength to your findings. It provides evidence that your results aren't driven by dichotomisation and misclassification problem raised by reviewer #1.

Response: Thank you for pointing this out; we now refer to Model S2c in the results section.

6. Second paragraph of discussion says that the findings "neither supported nor contradicted this hypothesis" which sounds odd. Surely it's one or the other?

Response: We agree and have now revised that sentence.

Minor housekeeping edits that the authors may wish to make:

- Typo in third paragraph of introduction: "decide which metric that it most suitable".
- In 'study sample' would confidentiality be a more appropriate word than "integrity"?
- Typo in patient and public involvement "advice" should be advise.

Response: These edits have been made.

- Would Table 3 be better suited to the supplementary material?

Response: We prefer to keep Table 3 in the main document, which is in accordance with the comments by Reviewer #1.

Tim Morris, University of Bristol

Reviewer: 1

Reviewer Name: Emily Murray

Institution and Country: University College London, United Kingdom Please state any competing interests or state 'None declared': None declared.

Please leave your comments for the authors below:

Thanks to the author(s) for addressing some of my previous comments. The manuscript has been improved on these points. Now that I have clarity on how the neighbourhood deprivation measure and exposure periods defined, I still find myself confused as to what models were being tested in the paper (see below) and whether these models match up with the stated aim of the paper. I believe the paper could still do with a re-write to clarify what life course models were tested (sensitive period, accumulation, trajectory, etc), how each models fits in with the research question and adding descriptors to results tables to link estimates back to the particular life course model being shown.

Response: We appreciate the previous comments by the Reviewer, which we believe have strengthened the manuscript, and apologize that certain unclarities still remained after our initial revision. The new comments on our revised manuscript by the Reviewer on how to improve the presentation of the different models were particularly valuable; when we read our manuscript again, we realized that the presentation of the models was unclear. The suggestions above were therefore very useful in order to make it easier for the potential readers. We have now clarified what models we used to examine our aims, which models that represent our sensitivity analyses and how these sensitivity analyses were used to evaluate our accumulation model. Please see the end of the introduction where our main aim is stated, i.e., to investigate if an accumulated exposure to neighborhood deprivation resulted in higher risks of CHD in addition to a more traditional single-point-in-time assessment. We have also revised the statistical analyses and added descriptors to the figures and tables to link our estimates to the models, as suggested above. In addition, we now list all the models at the end of the Methods section (please see below). This information can also be found in the supplementary material (Supplementary Table 1), together with the equations for each model described in detail together with their purpose (added in this revision, see comment below).

Summary of models

Model 1: Single-point-in-time model

Model 2: Categorical accumulated model

Model S2a: Sensitivity analysis A, Continuous accumulated model (five-year)

Model S2b: Sensitivity analysis B, Timing/period model

Model S2c: Sensitivity analysis C, Continuous accumulated model (one-year)

We have also clarified, at the end of the introduction, which life course model we tested. The following sentence is new:

“The accumulation hypothesis will be examined in the present study and represents one of several life course approaches in epidemiology that includes the study of long-term effects of different exposures on disease risk later in life.”

A. Previous #3. The overall aim of this study was to evaluate if there is an accumulated effect of exposure to neighbourhood deprivation on the risk of CHD. To construct a model that could be useful for this purpose, one necessary condition is that the model cannot assume what we intend to investigate (Kriebel, Checkoway et al. 2007, de Vocht, Burstyn et al. 2015). In other words, it is not possible to assume an increasing association by the number of years a person resides in a deprived neighbourhood and interpret an OR over 1 as a result of an accumulated effect.

I find the logic in this statement a bit puzzling. If your research question is whether the effect of neighbourhood deprivation on CHD accumulates over time/life course, then how will you test that without testing a model that assumes just that? From reading your paper, it is my understanding that you have assumed that exposure to neighbourhood deprivation is related to CHD occurs in different ways & then test which model has the best fit.

Response: We have deleted that statement, which also in accordance with the comments by Reviewer 2.

My understanding is that the models are as follows:

Model 1: Whether an individual has lived in a deprived neighbourhood at any time in each of the three five-year periods before the baseline. You call this a single-point-in-time measure, but I believe you have created more of a period prevalence measure. I would see single-point-in-time to be what the deprivation was in the neighbourhood they were residing at baseline. Or have I misunderstood the text in the measures section?

Response: Model 1 is based on the single-point-in-time measure for neighborhood deprivation and refers to baseline, which in this study represents the year before an individual attains the age when the five-year follow-up period starts. This has now been clarified in the methods section. The following sentence has been revised: “The exposure neighborhood variables used in the analyses were based either on a single-point-in-time assessment, assessed the year before baseline, or repeated assessments from the 15 years prior to baseline, divided into three five-year-periods.”

Model 2: The eight patterns of longitudinal exposure I would see as trajectory models, rather than strict accumulation.

Response: Model 2 represents an accumulated model with four exposure categories, i.e., never exposed [(0,0,0)], exposed during one of the three five-year periods [(1,0,0), (0,1,0) or (0,0,1)], exposed during two of the three five-year periods [(1,1,0), (0,1,1) or (1,0,1)], or exposed during all three five-year periods [(1,1,1)]. We have now clarified this in the methods section (see statistical analyses) and also added the following new sentence: "This model predicts CHD as a function of number of exposed periods without considering the timing of the exposure."

The accumulated model used in our study corresponds to the model described as "Accumulated model" in the paper by Mishra et al.

Model 2a: I would consider this model more in line with your stated research question. More time periods exposed equals more CHD risk.

Response: Model S2a represents one of our sensitivity analyses (in total 3, see above), which is an accumulated model where the exposure is examined in a continuous manner based on number of exposed five-year periods. Model 2 is our main model, which also is an accumulated model where the exposure is examined in a categorical manner. This has now been clarified in the methods section (see statistical analyses).

Model 2b: How is this different than model 2?

Response: Model S2b represents another of our sensitivity analyses that considers all eight exposure categories, taking the timing into account, in contrast to Model 2 where the timing is unimportant. This model corresponds to the saturated model in the paper by Mishra et al. and is included to evaluate whether the exposure has a similar effect if it occurs earlier or more recent. If exposure at earlier or more recent time-periods yields the same result, it is reasonable to use the simpler Model 2 rather than Model S2b. We evaluate this by comparing AIC values and this is in line with what is referred to as "Testing the accumulation hypothesis" in the paper by Mishra et al.

Model 2c: How does this differ from model 2a?

Response: Model S2c represents a sensitivity analyses, which is an accumulated model where the exposure is examined in a continuous manner based on exposed one-year periods. Model S2a is based on number of exposed five-year periods (see our response above).

Thank you for your questions about all the models. In our revisions, we have clarified which models that are used to examine the aims and which models that represent our sensitivity analyses. Please also see the added list “Summary of models”, just before the results section.

Suggestions:

1. Equations for each model would be extremely helpful for knowing exactly what you are modelling. There are some great papers, such as this one by Mishra et al (<https://academic.oup.com/ije/article/38/2/528/6555%2070>), where each model to be tested is stated explicitly. At the moment, I can't tell what was fitted for each model by the text alone.

Response: We have now added the equations for each model to the supplementary material (Supplementary Table 1) where we have listed the models in conjunction to the equations. The models are now also more explicitly described in the statistical analyses. Thank you for pointing out the useful paper by Mishra et al.

2. If your hypothesis is accumulation, then stick to testing accumulation models. If you're interested in whether the order/timing of when neighbourhood deprivation occurs, then test the mobility hypothesis (model 2 I believe). But if you include the latter, you need to add text to the introduction and discussion on justification for why you think timing of neighbourhood deprivation would matter for CHD risk.

Response: We apologize for being unclear. The focus of our paper was to investigate if an accumulated exposure to neighborhood deprivation resulted in higher risks of CHD in addition to a more traditional single-point-in-time assessment. Please see the end of the introduction where our main aim is stated. The timing assessments were only used in a sensitivity analysis (Model S2b). Please also see our response above where we have clarified how we have improved the presentation of our models.

3. It would also help if in the methods section you clearly defined what each model represented. Add these labels Table 3.

Response: We have now clarified what each model represents; please see the methods section (statistical analyses) and added the suggest labels to Table 3.

B. #5.4. Measures, page, line 45: are the four dimensions of deprivation defined somewhere? By 'definition' I meant how would you know a neighbourhood was classified as 'low income'? Was there a certain average neighbourhood income, certain percentage of households in the neighbourhood below a certain income threshold, etc?

Response: We have revised the second paragraph in the methods section with the subheading “Measures” in order to clarify that the measure was based on proportions of people residing in the neighborhood in question. We also refer to a paper that explains in more detail how the neighborhood deprivation score was constructed.

C. Supplementary tables: Why are only three estimates listed for models 2(crude), 2(adjusted) and S2a? There needs to be a lot more description on these tables to let the reader be able to understand what is being presented and results.

Response: We now realize that all headings for the supplementary tables were listed in the beginning of the document rather than in conjunction to the tables. We have now changed this and thank you for pointing this out. We have also added a footnote to supplementary Table 3a and Table 3b (former 2a and 2b) explaining how the presented ORs for Model S2a are derived and what they represent.

VERSION 3 – REVIEW

REVIEWER	Emily T Murray University College London United Kingdom
REVIEW RETURNED	09-Aug-2019

GENERAL COMMENTS	I want to commend the authors for the amount of work they have put into the revisions. The manuscript is now very clear in its methods, results and conclusions. The paper should be a valuable contribution to the neighbourhood effects field.
--

REVIEWER	Tim Morris University of Bristol, UK
REVIEW RETURNED	05-Aug-2019

GENERAL COMMENTS	I am satisfied with the revisions that the authors have made, which have clarified elements of the paper. Below are a couple of minor points and typos which the authors may wish to make. Page 12: “This also suggests that the changes in individual’s deprivation score over time is not driven by changes in neighborhood deprivation”. This is quite confusing and could probably be rephrased to make clearer. Do you mean that the changes for in deprivation weren’t driven by the same places changing over time? Page 13: I’m cautious about your statement “hypothesized causality in the association between neighborhood deprivation and CHD cannot be rejected”. This may be true, but more importantly it cannot be accepted either given your study design. Page 6 typo: New text reads “...exposure depend on at the year ...” Page 8 typo: “In a next sensitivity analysis, we constructed” Page 8 typo: “variables to explore on the possible effect”
--

VERSION 3 – AUTHOR RESPONSE

Reviewer(s)' Comments to Author:

Reviewer: 2

Reviewer Name: Tim Morris

Institution and Country: University of Bristol, UK Please state any competing interests or state ?None declared?: none declared

Please leave your comments for the authors below

I am satisfied with the revisions that the authors have made, which have clarified elements of the paper. Below are a couple of minor points and typos which the authors may wish to make.

Response: Thank you for your previous comments on the manuscript as well as pointing out some minor points and typos, which we now have corrected; please see below.

Page 12: ?This also suggests that the changes in individual?s deprivation score over time is not driven by changes in neighborhood deprivation?. This is quite confusing and could probably be rephrased to make clearer. Do you mean that the changes for in deprivation weren?t driven by the same places changing over time?

Response: Yes and thank you for noticing this. It has now been clarified.

Page 13: I?m cautious about your statement ?hypothesized causality in the association between neighborhood deprivation and CHD cannot be rejected?. This may be true, but more importantly it cannot be accepted either given your study design.

Response: We agree that the hypothesis cannot be accepted given our study design and have therefore modified that sentence. It now reads:

?Our results suggest that measures of accumulated exposure may be of greater importance in younger age cohorts and that a hypothesized causality in the association between neighborhood deprivation and CHD may be possible in younger but not in older age cohorts.?

Page 6 typo: New text reads ??exposure depend on at the year ??

Page 8 typo: ?In a next sensitivity analysis, we constructed?

Page 8 typo: ?variables to explore on the possible effect?

Response: Thank you; these typos have now been corrected.

Reviewer: 1

Reviewer Name: Emily T Murray

Institution and Country: University College London United Kingdom

Please state any competing interests or state ?None declared?: None declared.

Please leave your comments for the authors below

I want to commend the authors for the amount of work they have put into the revisions. The manuscript is now very clear in its methods, results and conclusions. The paper should be a valuable contribution to the neighbourhood effects field.

Response: Thank you, we are very pleased to hear that the manuscript is now clearer and will make a valuable contribution to the research field.

VERSION 4 – REVIEW

REVIEWER	Tim Morris University of Bristol, UK
REVIEW RETURNED	16-Aug-2019
GENERAL COMMENTS	Great job, and congratulations on the paper!